# Prognostic Implications of 18-FDG Positron Emission Tomography/Computed Tomography in Resectable Pancreatic Cancer

**DOI:** 10.3390/jcm9072169

**Published:** 2020-07-09

**Authors:** Cosimo Sperti, Alberto Friziero, Simone Serafini, Sergio Bissoli, Alberto Ponzoni, Andrea Grego, Emanuele Grego, Lucia Moletta

**Affiliations:** 1Department of Surgery, Oncology and Gastroenterology, 3rd Surgical Clinic, University of Padua, Via Giustiniani 2, 35128 Padua, Italy; alberto.friziero@unipd.it (A.F.); simone.serafini@ymail.com (S.S.); andrea.grego@studenti.unipd.it (A.G.); lucia.moletta@unipd.it (L.M.); 2Nuclear Medicine, Belluno General Hospital, Viale Europa 22, 32100 Belluno, Italy; sergiocelico@gmail.com; 3Department of Radiology, Padua General Hospital, Via Giustiniani 2, 35128 Padua, Italy; alberto.ponzoni@aopd.veneto.it; 4University of Padua, Via Giustiniani 2, 35128 Padua, Italy; emanuele.grego@studenti.unipd.it

**Keywords:** fluorodeoxyglucose, pancreatectomy, pancreatic cancer, positron emission tomography, prognosis, standardized uptake value

## Abstract

There are currently no known preoperative factors for determining the prognosis in pancreatic cancer. The aim of this study was to examine the role of 18-fluorodeoxyglucose (18-FDG) positron emission tomography/computed tomography (18-FDG-PET/CT) as a prognostic factor for patients with resectable pancreatic cancer. Data were obtained from a retrospective analysis of patients who had a preoperative PET scan and then underwent pancreatic resection from January 2007 to December 2015. The maximum standardized uptake value (SUVmax) of 18-FDG-PET/CT was calculated. Patients were divided into high (>3.65) and low (≤3.65) SUVmax groups, and compared in terms of their TNM classification (Union for International Cancer Contro classification), pathological grade, surgical treatment, state of resection margins, lymph node involvement, age, sex, diabetes and serum Carbohydrate Antigen 19-9 (CA 19-9) levels. The study involved 144 patients, 82 with high SUVmax pancreatic cancer and 62 with low SUVmax disease. The two groups’ disease-free and overall survival rates were significantly influenced by tumor stage, lymph node involvement, pathological grade, resection margins and SUVmax. Patients with an SUVmax ≤ 3.65 had a significantly better survival than those with SUVmax > 3.65 (*p* < 0.001). The same variables were independent predictors of survival on multivariate analysis. The SUVmax calculated with 18-FDG-PET/CT is an important prognostic factor for patients with pancreatic cancer, and may be useful in decisions concerning patients’ therapeutic management.

## 1. Introduction

Pancreatic cancer is only the 12th most common cancer worldwide, but it is the 7th most common cause of cancer-related death [1]. The number of new cases of pancreatic cancer will continue to rise in future, largely due to population aging and growth. In the United States, pancreatic cancer was the second most common gastrointestinal malignancy in 2018 [2]. In the European Union (EU), it was estimated that deaths from pancreatic cancer surpassed those due to breast cancer in 2017, making the disease the third most important cause of cancer-related death in the EU, after lung and colorectal cancer [3]. The prognosis for pancreatic cancer is generally poor, with five-year survival rates in the range of 6% to 10% [4,5]. Approximately 80% of patients have regional spread or metastatic disease at the time of their diagnosis. Hence the need for enhanced screening modalities, early detection, accurate preoperative staging, and improved treatment options. Surgery is the only potentially curative treatment for pancreatic cancer [6]. Unfortunately, only 15–20% of patients are candidates for pancreatectomy due to the above-mentioned high proportion of cases of advanced disease at presentation. Neoadjuvant therapy, defined as treatment (chemotherapy and/or radiation) administered prior to surgery, has been advocated for locally-advanced pancreatic cancer, and also for potentially resectable disease. The possible benefits lie in earlier treatment reducing the likelihood of distant disease, and tumor downstaging optimizing resection. The potential problems associated with neoadjuvant treatments for resectable tumors concern the risk of the cancer progressing to unresectable tumor during such therapy, the differences in the neoadjuvant therapy protocols adopted at different centers, and the current lack of strong evidence to support its efficacy [7].

Many circulating, molecular and clinicopathological factors have been thoroughly investigated in efforts to predict the survival of patients with pancreatic cancer [8]. Attention has focused especially on tumor stage and pathological grade [9,10], resection margins [11], preoperative serum CA 19-9 levels, postoperative normalization of tumor markers [12,13], and the demonstration of disseminated tumor cells [14]. There have been conflicting results, however, and a different survival for patients with the same stage of disease is not infrequent. 

18-fluorodeoxyglucose positron emission tomography (18-FDG-PET) is a noninvasive imaging technique based on the principle of specific tissue metabolism, with a selective 18-FDG uptake and retention by malignant cells [15,16]. PET has been proposed for diagnosing and staging various malignancies, including pancreatic carcinoma [17,18]. There is evidence of 18-FDG uptake in malignant tumors being related to a tumor’s aggressiveness. Some authors [19,20,21] have reported on prognostic information obtained with 18-FDG-PET in patients with pancreatic cancer, or outlined the role of PET in predicting early recurrences after surgery [22,23,24]. The numbers of patients included in these studies were too small to draw any final conclusions, however. This is particularly true for patients with localized pancreatic cancer (stage I–II) amenable to resection as part of a multimodality approach to pancreatic adenocarcinoma. We previously found PET an independent prognostic marker in a series of patients with pancreatic cancer, including a small subset of resectable tumors (*n* = 16) [25].

The aim of the present study on a series of patients with resectable pancreatic adenocarcinoma was to ascertain whether glucose metabolism, as assessed with 18-fluorodeoxyglucose positron emission tomography/computed tomography (PET/CT), provides additional prognostic information, over and above the established prognostic factors, in patients with pancreatic cancer.

## 2. Materials and Methods

### 2.1. Patients Population

Data were obtained from a retrospective analysis of a prospective database of patients who underwent pancreatic resection at our department between January 2007 and December 2015. Patients with intraductal papillary mucinous neoplasms, endocrine tumors, cystic neoplasms, pancreatic metastases, and duodenal, ampullary and bile duct cancers were excluded. This left 144 consecutive patients with pancreatic cancer who had a PET/CT scan (with semiquantitative analysis of the tracer uptake) as part of their preoperative work-up within 30 days before undergoing pancreatic resection, who were enrolled in the present study. Informed consent was obtained from each patient.

The sample was a mean 66.3 years old (range 48–82), and consisted of 70 males and 74 females. Pancreatic ductal adenocarcinoma was confirmed at histology on surgical specimens in all cases. All resection procedures were performed by the same surgical team. A limited involvement of the superior mesenteric-portal axis (less than 2 cm) in the absence of extrapancreatic disease, or involvement of the superior mesenteric artery and/or celiac trunk, were not considered as contraindications to surgery. None of the patients received neoadjuvant therapies. Resection of the pancreas entailed pylorus-preserving pancreaticoduodenectomy (PD) for tumors of the head of the pancreas, and distal pancreatectomy with splenectomy for tumors of the body and tail. Total pancreatectomy was reserved for cases where the resection margin of the pancreas was involved by the tumor or when a pancreatic anastomosis was judged at high risk of leakage. All patients underwent standard lymph node dissection—5, 6, 8a, 12b1, 12b2, 12c, 13a, 13b, 14a and 14b right lateral side, 17a, 17b, [26] and para-aortic node sampling for pancreatic head carcinoma, and 8a, 9,10,11, 18 for patients with pancreatic body and tail cancers. Para-aortic nodes were excised by harvesting the lymphocellular aortocaval tissue located below the left renal vein up to the origin of the inferior mesenteric artery (station 16b1). Resections were defined as curative (R0) when the pathology report confirmed negative resection margins, or R1 in the presence of tumor ≤ 1 mm from the resection margins, according to Leeds criteria [27]. Tumors were staged according to the Union for International Cancer Control (UICC)TNM classification [28]. Each patient’s clinical and pathological records were reviewed, and the following characteristics were included in our analysis—age, sex, diabetes, type of surgery, preoperative serum CA 19-9 levels (RIA, Centocor Inc., Malvern, PA, reference: < 37 kU/L), tumor stage, lymph node status, pathological grade, R0 resection, disease-free survival and overall survival. Disease-free survival (DFS) was measured from the date of surgery to the date of radiologically detected recurrence or censoring. Overall survival (OS) was measured from the date of surgery to the date of death or censoring. All patients underwent regular follow-up, which included a physical examination, abdominal CT or US, and tumor marker assay every 3 months for the first 2 years, and every 6 months thereafter. Adjuvant gemcitabine-based chemotherapy was scheduled for all patients, whenever applicable.

Ethical approval: All procedures performed in studies involving human participants were in accordance with the ethical standards of the institutional and/or national research committee and with the 1964 Helsinki Declaration and its later amendments or comparable ethical standards.

### 2.2. 18-FDG-PET/CT Imaging

18-FDG-PET/CT images were obtained using two different dedicated tomographs—Biograph-16™, Siemens Healthcare GmbH, Erlangen-Germany from 2007 to 2012, and Discovery™, GE-Healthcare, Boston USA in the years 2013–2015.

Each scan was performed 50–70 min postinjection of 150–400 MBq of FDG in fasting patients (almost 6 hours), with serum levels of glucose < 110 mg/dL for nondiabetic patients and < 200 mg/dL in diabetic ones; in order to avoid interferences due to hyperglycaemia, blood glucose was checked just before the procedure.

The acquisitions were performed with standard modalities (scan length from base skull to 1/3 prox of legs, 6–7 beds, 2–3 min/bed); when necessary, a limited second scan of 2 beds with the same modalities was repeated on the hepato-pancreatic region at 90-100 min postinjection. 

Images were reconstructed with standard algorithms, and the SUV value was calculated in the suspected neoplastic foci (SUV = tissue tracer concentration/injected dose/body weight); for the SUV analysis, a circular region of interest was placed over the area of maximal focal FDG uptake suspected to be a tumorous focus (SUVmax).

After acquisition, scan images were interpreted and referred by an experienced Nuclear Medicine physician, well-trained in PET/TC (almost five years). 

### 2.3. Statistical Analyses

Statistical analyses were run using STATA, version 14.1 (4905 Lakeway Drive College Station, Texas, 77845, USA). Receiver operating characteristic (ROC) curve analysis was used to ascertain the optimal cut-off for predicting DFS and OS after pancreatectomy. The optimal cut-off was identified as the point of intersection nearest the top left-hand corner between the ROC curve and the diagonal line from the top right-hand corner to the bottom left-hand corner of the graph. For the univariate analysis, the patients were divided into two groups, with SUVmax (> vs. ≤ 3.65) as the cut-off. Differences between the characteristics of the patients in the two groups were tested for significance using the Mann–Whitney U test, chi-square test, Fisher’s exact test or t-student test. Univariate and multivariate analysis were used to investigate the effect of the following variables on survival—age, sex, tumor stage, pathological grade, lymph node involvement, resection margins, diabetes, and serum CA 19-9 levels. Survival data were estimated with the Kaplan–Meier method and examined using the log-rank test. Multivariate analysis of survival was performed using Cox’s proportional hazards model. Significance was set at *p* < 0.05.

## 3. Results

Table 1 shows the clinical and pathological details of the 144 patients. Fifty-three patients had diabetes, and 93 had preoperative serum CA 19-9 levels above the normal limit. The surgical procedure involved pylorus-preserving PD in 106 patients, distal pancreatectomy with splenectomy in 34, and total pancreatectomy in four. A segmental portal-mesenteric vein resection was included in 21 cases. The resection margins were positive (R1) in 38 patients (26.4%). Lymph node metastases (stage II1) were identified in 103 patients, 114 had stage I-II tumor, and 95 tumors (66%) were well- or moderately-differentiated. A total of 132 patients (92%) received gemcitabine-based adjuvant chemotherapy.

The median SUVmax of the 144 patients was 4.0 (range 1.0 to 12.0). From the ROC analysis, the best cut-off was identified at 3.65. The area under the ROC curve (AUC) was 0.66 (95%CI 0.542–0.77) (Figure 1). When patients were grouped by low SUVmax (≤ 3.65) versus high SUVmax (> 3.65), the two groups did not differ statistically in terms of age, sex, number of patients with of diabetes, tumor stage, type of treatment, or number of patients given adjuvant therapy. Median values of CA 19-9, numbers of patients with lymph node metastases and those with poorly-differentiated tumors were significantly higher in the high SUVmax group (Table 1). The 144 patients’ median serum CA 19-9 level was 114 kU/L (range 1.0 to 6637 kU/L). Follow-up was available for all patients, and ranged from 6 to 152 months.

### 3.1. Disease-Free Survival

With a median follow-up of 56.7 months (range 2–70), pancreatic cancer recurred in 126/144 patients (87.5%). The median DFS was 11.6 months. 

On univariate Cox regression analysis (Table 2), lymph node metastases, pathological grade, resection margins, tumor stage, and SUVmax correlated significantly with DFS, while diabetes and serum CA 19-9 levels did not. Multivariate Cox regression analysis (Table 2) showed that the same parameters were independent predictors of DFS. Patients with a preoperative SUVmax > 3.65 had a significantly shorter DFS than patients with a SUVmax ≤ 3.65 (*p* = 0.001) (Figure 2). When the patients grouped by SUVmax were stratified by stage of disease, 18-FDG-PET/CT uptake correlated with survival even among patients in stage I-II, with a better survival for patients with SUVmax ≤3.65 (*p* = 0.0004) (Figure 3)

### 3.2. Overall Survival 

With a median follow-up of 100.8 months (range 6–152), 125/144 patients (87%) died of pancreatic cancer, and another two patients died of causes unrelated to their pancreatic disease. The median OS for the whole cohort was 22.4 months (range 19–27). At univariate Cox regression analysis (Table 3) lymph node metastases, pathological grade, resection margins, tumor stage, and SUVmax correlated significantly with OS. Multivariate Cox regression analysis (Table 3) identified the same variables as being significantly associated with OS. As in the case of DFS, diabetes and CA 19-9 serum levels were not independent predictors of OS. Survival analysis with the Kaplan–Meier method showed a significantly lower OS for patients with a preoperative SUVmax > 3.65 than for those with a SUVmax ≤ 3.65 (*p* < 0.001) (Figure 4). When the patients grouped by SUVmax were stratified by tumor stage, 18-FDG uptake correlated with OS among patients with stage I-II (better survival for patients with SUVmax ≤ 3.65, *p* = 0.0002), but not for those with stage III–IV tumors (*p* = 0.71). The survival curves for patients with stage I–II and SUVmax > 3.65 did not differ statistically from those of patients with stage III–IV and SUVmax ≤ 3.65. (Figure 5). At latest follow-up, 17 patients were alive (16 disease-free): 13 in the group with SUVmax ≤ 3.65, and 4 in the group with SUVmax > 3.65 (one with recurrent cancer). 

## 4. Discussion

An accurate pretreatment prognosis for patients with pancreatic cancer would be very helpful for the purpose of tailoring their treatment (either surgery or multimodality clinical management). This is particularly true for apparently localized, resectable carcinoma of the pancreas because several authors have recommended neoadjuvant therapy for such patients rather than upfront surgery, the benefits of which have yet to be definitely established. The rationale for using PET/CT preoperatively for prognostic purposes in cases of pancreatic cancer stems from evidence of an accelerated glucose transport rate and increased rate of glycolysis being among the most characteristic biochemical markers of malignant transformation. Overexpression of glucose transporter 1 (Glut-1) [29] and glycolytic enzymes [30] has been demonstrated in human pancreatic adenocarcinoma. 18-FDG is a glucose analog actively taken up into the pancreatic cell by Glut-1 and phosphorylated by hexokinase in the first step of glycolysis. Its accumulation thus reflects the rate of carbohydrate metabolism and the malignant activity of a pancreatic cancer [22]. The standardized uptake value (SUV), a semiquantitative parameter of glucose consumption that enables a quantitative estimation of 18-FDG accumulation, can easily be obtained preoperatively on PET/CT. 18-FDG-PET/CT is therefore useful for distinguishing benign from malignant tumors, for diagnosing tumor recurrences, and for assessing the effects of neoadjuvant chemoradiation therapies [20,31,32]. Preliminary evidence of the correlation between 18-FDG uptake and prognosis for pancreatic adenocarcinoma has been reported in small series of patients [21,25,33]. Nakata et al. [33] introduced 18-FDG PET and SUV as metabolic prognostic factors in patients with pancreatic carcinoma. In a small series of 14 patients, they found survival significantly shorter in the high SUV group (>3.0) than in the low SUV group (<3.0) (*p* < 0.05). These results were only partially confirmed, however, by the same authors four years later [19] in 37 patients with histologically-confirmed pancreatic cancer. While SUV was unable to predict survival for patients with resectable tumor, among those with unresectable disease, patients with a low SUV survived significantly longer than those with a high SUV (*p* = 0.03); furthermore, multivariate analysis confirmed tumor SUV as an independent prognostic indicator for patients with unresectable tumors.

In the present study, we analyzed 18-FDG uptake in a cohort of patients (*n* = 144) with histologically-confirmed pancreatic cancer. When grouped by high (> 3.65) versus low (≤ 3.65) SUVmax, patients did not differ statistically in terms of age, sex, tumor stage, pathological grade, serum CA 19-9 levels, diabetes, or type of treatment. DFS and OS were significantly influenced by SUVmax, however, being 20 and 28 months, respectively, for low-SUVmax patients as opposed to 9 and 19 months for high-SUVmax patients (*p* = 0.001). Among the clinicopathological variables considered, tumor stage, pathological grade, lymph node involvement, and resection margins correlated significantly with both DFS and OS after univariate analysis. Multivariate analysis confirmed SUVmax, tumor stage, grade, resection margins and lymph node status as independent predictors of DFS and OS.

Interestingly, when patients in the two SUVmax groups were stratified by tumor stage, 18-FDG uptake significantly influenced survival for cases in stage I–II, but not for those in stage III–IV. Serum CA 19-9 levels and diabetes had no influence on survival. The different biological aggressiveness of the tumor indicated by the SUVmax may explain the different survival rates after potentially curative resection with otherwise similar prognostic variables. 18-FDG-PET/CT is known to be a less accurate indicator in diabetic patients, and may be unable to predict their survival adequately. Some authors [34,35] recently reported that preoperative SUVmax and serum CA 19-9 independently predicted pathological stages and OS. However, it is hard to establish an optimal cut-off value of CA 19-9 as a reproducible preoperative prognostic factor, because 10–15% of the population does not express CA 19-9 and because the levels of such tumor markers are notoriously influenced by liver and renal insufficiency [36].

Our results confirm previous evidence [22,23,37,38,39,40,41,42] of SUVmax (measured in terms of the tumor’s uptake of 18-FDG) being a simple and reliable pretreatment prognostic parameter, as in other malignancies. A summary of the results of other studies on SUVmax as a prognostic factor in cases of resectable pancreatic cancer is given in Table 4.

Including ours, seven studies have been published [22,23,38,39,40,41], all retrospective, concerning a total of 658 patients. SUVmax cut-offs vary greatly, but all the studies report a significantly longer DFS for patients with a low SUVmax, and 4 studies also describe a significantly better OS [23,38,40,41]. Since SUVmax only gives an indication of peak metabolic activity, not of tumor burden, some authors have explored the value of metabolic tumor volume (MTV) and total lesion glycolysis (TLG) as predictors of pancreatic cancer outcome [40,43,44]. Xu et al. [44] found that MTV and TLG independently predicted OS and DFS, and did so better than CA 19-9 levels, SUVmax, or tumor size. These findings were confirmed by Lee et al. [40] in 87 patients with resectable carcinoma of the pancreas (30 treated with neoadjuvant therapy)—MTV and TLG were independent prognostic factors irrespective of neoadjuvant therapy. On the other hand, SUVmax is less time-consuming and easier to calculate, and in our and others’ experience, it provides the same important information. 

Several previous studies found the tumor’s histological characteristics important in establishing the prognosis for pancreatic cancer patients [8,9,10,11,12,13,14], but most of them are only available after surgery. The great advantage of the SUVmax calculated on 18-FDG-PET/CT is that it can be obtained before any treatment is undertaken. As the prognostic value of SUVmax is equivalent to that of tumor staging, stratifying patients by extent of disease on multidetector CT scans and SUVmax may improve our understanding of the actual effect of different treatments. 

There is evidence to suggest that glycolytic activity as measured from 18-FDG uptake gives an indication of a tumor’s growth and biological behavior, enabling a prediction of patients’ DFS and OS. 18-FDG-PET/CT might therefore be used to identify patients with resectable pancreatic cancer at higher risk of early recurrence and shorter survival who could benefit from neoadjuvant therapy. The feasibility and clinical usefulness of this approach would need to be confirmed in prospective trials.

Another topic of interest could be the evaluation of SUVmax with 18-FDG PET/CT measured before and after chemotherapy in those patients scheduled for neoadjuvant therapy and its association with their survival. 

There are some limitations of our study to mention. First, it was a retrospective study conducted at a single institution. Second, various drugs were used for adjuvant therapy during the study period, and this may have influenced the results. The significant number of patients and PET findings considered nonetheless sufficed to show statistically significant and clinically relevant differences.

## 5. Conclusions

The SUVmax calculated on 18-FDG-PET/CT provides useful prognostic informations in patients with pancreatic cancer before any surgical or medical treatment is administered, and may therefore help stratify patients for prospective studies comparing different treatment options (surgery versus chemotherapy). 

## Figures and Tables

**Figure 1 jcm-09-02169-f001:**
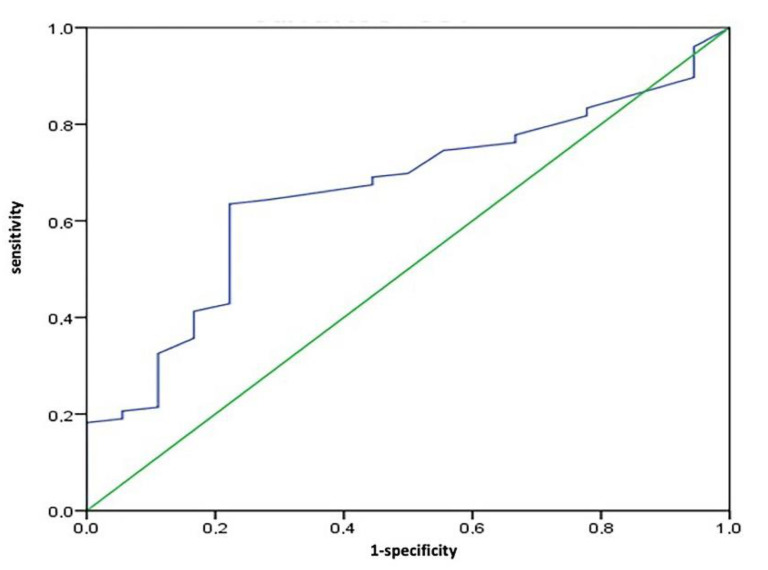
Receiver operator characteristic (ROC) curve for maximum standardized uptake value (SUVmax) cut-off, showing that the most effective cut-off was 3.65 (AUC 0.659, 95%CI 0.542–0.77).

**Figure 2 jcm-09-02169-f002:**
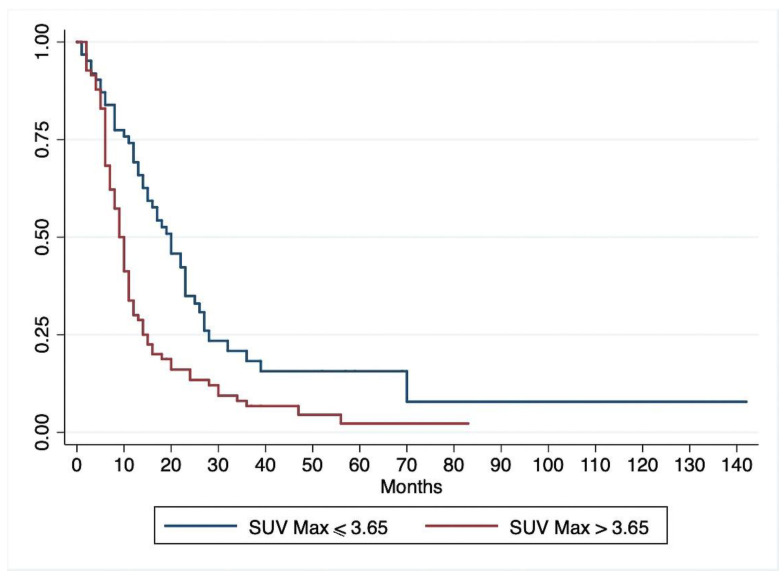
Kaplan–Meier curve for disease-free survival estimated for patients with preoperative SUVmax > 3.65 and those with SUVmax ≤ 3.65.

**Figure 3 jcm-09-02169-f003:**
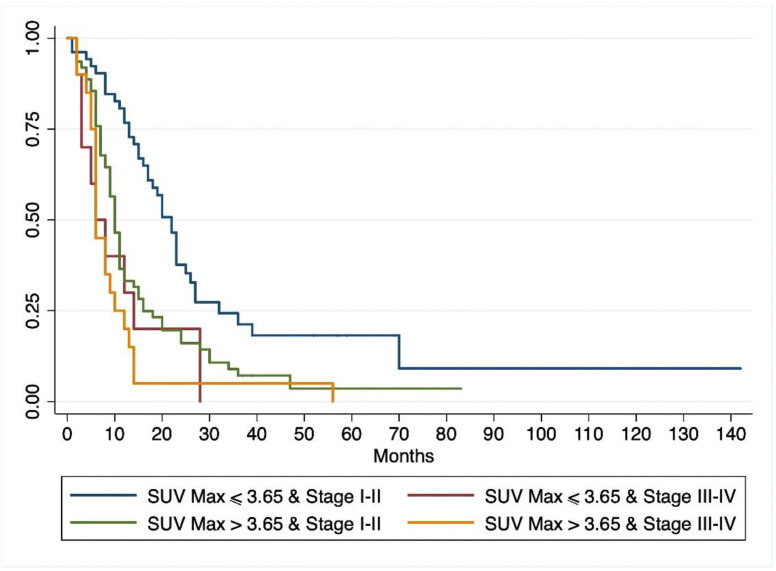
Kaplan–Meyer estimates for disease-free survival based on preoperative tumor stage and high or low SUVmax.

**Figure 4 jcm-09-02169-f004:**
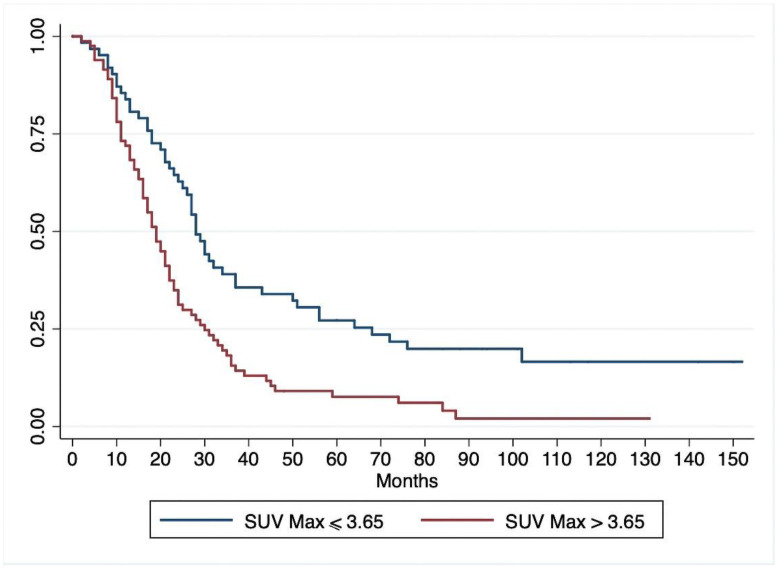
Kaplan–Meier curves for overall survival of patients with preoperative SUVmax > 3.65 and those with SUVmax ≤ 3.65.

**Figure 5 jcm-09-02169-f005:**
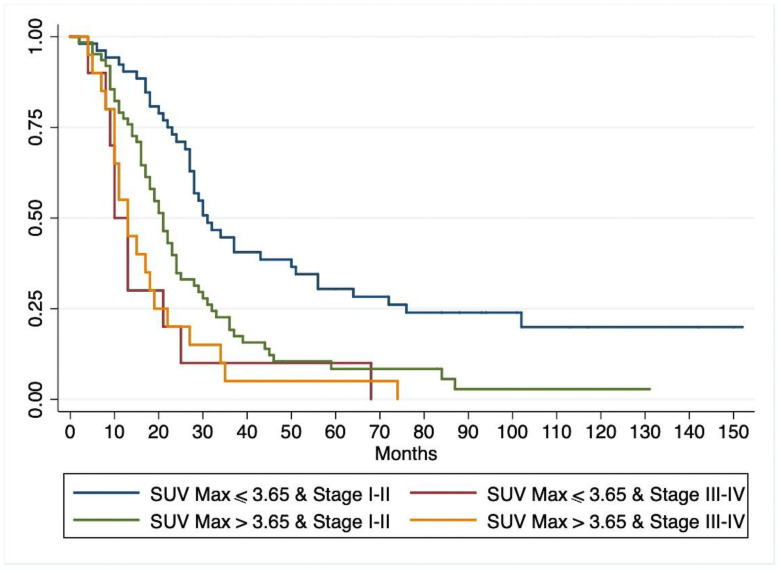
Kaplan-Meyer curves for overall patient survival by preoperative stage and SUVmax category.

**Table 1 jcm-09-02169-t001:** Standardized Uptake Values and Patients’ Clinical and Pathological Details.

	All Patients	SUVmax ≤ 3.65	SUVmax > 3.65	*p* Value
Patients, *n* (%)	144	62 (43.1%)	82 (56.9%)	
Age, yrs (mean ± SD)	66.32 ± 11.40	66.48 ± 09.32.	67.55 ± 10.31	
Sex M	70	32	38	
F	74	30	44	
UICC				0.158
I–II, *n* (%)	114 (79.2%)	52 (45.6%)	62 (54.4%)	
III–IV, *n* (%)	30 (20.8%)	10 (33.3%)	20 (66.7%)	
Grade, *n* (%)				0.023
Well- or moderately differentiated (G1–G2)	95 (66%)	47 (49.5%)	48 (50.5%)	
Poorly-differentiated (G3)	49 (34%)	15 (30.6%)	34 (69.4%)	
Resection margins				0.232
R0, *n* (%)	106 (73.6%)	48 (45.3%)	58 (54.7%)	
R1, *n* (%)	38 (26.4%)	14 (36.8%)	24 (63.2%)	
Lymph nodes				0.036
Negative, *n* (%)	41 (28.5%)	23 (56.1%)	18 (43.9%)	
Positive, *n* (%)	103 (71.5%)	39 (37.9%)	64 (62.1%)	
Diabetes				0.170
No, *n* (%)	90 (62.5%)	42(46.7%)	48(53.3%)	
Yes, *n* (%)	54 (37.5%)	20 (37%)	34 (63%)	
SUVmax, mean (±SD)	5 (±3.2)	2.6 (±1.2)	6.9 (±3.1)	
Serum CA 19-9, mean (±SD)	524.5 (±1123)	392.9 (±1051.9)	623.9 (±1172.1)	0.88
Serum CA 19-9, median (IQR), range	114 (IQR 23–382) range 1–6637	52.9 (IQR 18–256) range 1–6637	154.35 (IQR 27–470) range 1–5460	0.032
CA 19-9 < 114 kU/L	81 (56.3%)	41 (50.6%)	40 (49.4%)	0.028
CA 19-9 > 114 kU/L	63 (43.7%)	21 (33.3%)	42 (66.7%)	
OS, median (95%CI)	22 (19–27)	28 (24–37)	19 (16–22)	0.002
DFS, median (95%CI)	12 (10–14)	20 (14–23)	9 (8–11)	0.001

**Table 2 jcm-09-02169-t002:** Association Between Preoperative Variables and Disease-Free Survival on Univariate ^a^ and Multivariate ^b^ Cox Regression Model. HR = hazard ratio.

Variables	HR ^a^	95%CI ^a^	*P* Value ^a^	HR ^b^	95%CI ^b^	*P* Value ^b^
Lymph node metastases	2.33	1.511–3.596	<0.0001	1.779	1.130–2.800	0.013
Pathological grade	1.581	1.090–2.293	0.016	1.661	1.137–2.426	0.009
Radicality	2.047	1.377–3.044	<0.0001	1.840	1.223–2.769	0.003
Stage	2.181	1.429–3.330	<0.0001	1.787	1.144–2.794	0.011
Diabetes	1.352	0.942–1.941	0.102	-	-	-
SUVmax	1.106	1.051–1.165	<0.0001	1.085	1.025–1.148	0.004
CA 19-9	1.001	0.999–1.001	0.312	-	-	-

**Table 3 jcm-09-02169-t003:** Association Between Preoperative Variables and Overall Survival on Univariate ^a^ and Multivariate ^b^ Cox Regression Model.

Variables	HR ^a^	95%CI ^a^	*P* Value ^a^	HR ^b^	95%CI ^b^	*P* Value ^b^
Lymph node metastases	2.433	1.588–3.721	<0.0001	1.730	1.101–2.719	0.017
Pathological grade	1.493	1.030–2.165	0.034	1.484	1.017–2.163	0.040
Radicality	2.352	1.583–3.495	<0.0001	2.079	1.374–3.147	0.001
Tumor stage	2.489	1.637–3.784	<0.0001	2.127	1.369–3.305	0.001
Diabetes	1.222	0.851–1.756	0.278	-	-	-
SUVmax	1.074	1.025–1.124	0.002	1.055	1.001–1.111	0.044
CA 19-9	1.001	0.999–1.001	0.196	-	-	-

**Table 4 jcm-09-02169-t004:** The Literature Reporting Differences in Overall Survival and Disease-Free Survival by SUVmax.

Author	Year	Design	*n*	SUVmax	OS (mo)	*p*	DFS (mo)	*p*
Okamoto et al. [22]	2011	R	56	<5.5 >5.5	NA	-	NA	0.025
Choi et al. [38]	2013	R	64	≤3.5 >3.5	45.4 vs. 23.5	0.011	26.1 vs. 9.2	0.002
Lee et al. [40]	2014	R	87	≤4.7 >4.7	34.4 vs. 20.6	0.03	12.9 vs. 9.9	0.03
Kitasato et al. [39]	2014	R	41	≤3.4 >3.4	NR	-	610 vs. 354 days	0.04
Yamamoto et al. [23]	2015	R	128	<6.0 ≥6.0	37 vs. 18	<0.001	23 vs. 6	<0.001
Ariake et al. [41]	2018	R	138	<4.85 ≥4.85	50.4 vs. 21.5	<0.001	24.3 vs. 10.3	<0.001
Present series	2020	R	144	≤3.65 >3.65		<0.001		<0.001

R = retrospective; OS = overall survival; DFS = disease-free survival; NR = not reported; NA =not applicable; mo = months.

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
