# Peer review of "Prognostic Implications of 18-FDG Positron Emission Tomography/Computed Tomography in Resectable Pancreatic Cancer"

_jcm, 2020, doi:10.3390/jcm9072169_

Round 1

Reviewer 1 Report

The authors examine the role of 18 FDG PET/CT as a prognostic factor in pancreatic cancer patients. They retrospectively looked at 144 patients who had a PET/CT prior to surgery and did not have neoadjuvant therapy. They concluded that patients with a higher SUVmax had worse DFS and OS and suggested that this approach could be useful to make treatment decisions (surgery vs. neoadjuvant therapy) based on SUVmax cutoff. 

Though it is very important to develop prognostic factors to ultimately stratify patients to treatment options, applying this method to additional cohorts and incorporation of biomarkers (NGS from surgical sample, etc) should be considered before such a strong conclusion. One such cohort of patients to consider looking into are those who had PET/CT before and after neoadjuvant therapy. This could be done retrospectively if the cohort is available or can be done in a prospective manner. Additional cohorts may be beyond the scope of this specific study, however, the authors should consider mentioning how their prognostic PET/CT approach could be further testing in a population who undergo neoadjuvant therapy. 

A cohort of 144 patients is not considered large, the authors mention the word "large" multiple times. I would suggest removing the word "large" as it is very subjective. 

Author Response

  • Observation 1:

Though it is very important to develop prognostic factors to ultimately stratify patients to treatment options, applying this method to additional cohorts and incorporation of biomarkers (NGS from surgical sample, etc) should be considered before such a strong conclusion. One such cohort of patients to consider looking into are those who had PET/CT before and after neoadjuvant therapy. This could be done retrospectively if the cohort is available or can be done in a prospective manner. Additional cohorts may be beyond the scope of this specific study, however, the authors should consider mentioning how their prognostic PET/CT approach could be further testing in a population who undergo neoadjuvant therapy.

Reply:

Your observation that 18 FDG PET/CT could not be the only predictor to orientate surgical vs neoadjuvant approach in pancreatic neoplasm is agreeable. In particular, Next Generation Sequencing (NGS) from biopsy will probably provide a personalized approach to choose neoadjuvant/adjuvant therapy or target therapy. Nevertheless, there is a tumor heterogeneity in the genetic profile of pancreatic cancer; impact of neoadjuvant therapy on various genetic profile is under-evaluation and in this situation is an onerous method to make decision. Probably in the future NGS will be relevant and guide clinical practice. And we’ll incorporate in our next study.

Regards the suggestion of making PET/CT before and after neoadjuvant chemotherapy, we stated in Material and Methods that our patients had resectable pancreatic cancer that didn’t undergo neoadjuvant chemotherapy. Your proposal of extending this protocol to investigate changes in pre- and post- neoadjuvant-chemotherapy patients is very interesting. We certainly intend to investigate this association in future prospective study. Thank a lot for your advice. (we add into discussion a proposal of next study: lines 312-314)

  • Observation 2:

A cohort of 144 patients is not considered large, the authors mention the word "large" multiple times. I would suggest removing the word "large" as it is very subjective.

Reply:

we remove the word “large” from the text (Modified lines 80; 261, 318).

Reviewer 2 Report

This manuscript details the retrospective analysis of 144 patients who had a preoperative PET scan followed by pancreatectomy for pancreatic cancer. They concluded that high SUVmax is an independent predictor of poor survival and may be useful in patient stratification in prospective studies comparing different treatment options. This paper is well-written and interesting. However, I have a couple of major concerns.

  1. This study includes patients who underwent pancreatectomy between 2009 and 2015. However, the authors note that the follow-up ranged from 6 to 180 months (15 years). Further, Kaplan-Meier curves shown in figures 2-5 implicate that this cohort includes patients who survived longer than 140 or 150 months despite being 2020 now. This may bring to cause doubts about data credibility and accuracy. The authors should review the database to do statistical analysis.
  2. As cited in references (Yamamoto et al. and Ariake et al.), the prognostic impact of SUVmax has been reported previously. This study is well-conducted analyzing a large number of patients. Results however only confirm previous reports, yet do not provide novel hypothesis-generating insights.

Minor comments:

  1. The authors should clearly state in abstract that patients with HIGH SUVmax have significantly WORSE survival compared to those with low SUVmax.
  2. What kind of tomograph was used in 2009?
  3. It is noted that there were 39 patients with R1 resection in main manuscript, whereas 38 patients in Table 1.
  4. P values for serum CA 19-9 should be shown in Table 1.
  5. It would be recommended to do multivariate analysis using Cox proportional hazards model but not logistic regression analysis.

Author Response

Major Concern

Observation 1:

  1. This study includes patients who underwent pancreatectomy between 2009 and 2015. However, the authors note that the follow-up ranged from 6 to 180 months (15 years). Further, Kaplan-Meier curves shown in figures 2-5 implicate that this cohort includes patients who survived longer than 140 or 150 months despite being 2020 now. This may bring to cause doubts about data credibility and accuracy. The authors should review the database to do statistical analysis.

Reply:

Thank you for your revision. We apologized , this is a transcription error we wrote “2009” instead of “2007” and “180” instead of “152”.

In fact: Our patients underwent pancreatectomy between January 2007 and December 2015 and the range of follow-up is 6 to 152 months.

In 2007 there are 20 patients with a diagnosis of pancreatic cancer, of these two long survivors had follow-up of 150 and 152 respectively. (Modified lines 26, 88, 173, 208)

Then the survival analysis was correctly conducted together with Kaplan-Meier, log-rank and Cox regression model.

Observation 2:

  1. As cited in references (Yamamoto et al. and Ariake et al.), the prognostic impact of SUVmax has been reported previously. This study is well-conducted analyzing a large number of patients. Results however only confirm previous reports, yet do not provide novel hypothesis-generating insights.

Reply:

We agree that prognostic impact of SUVmax has been reported previously, but cut-off values in different series are quite variable and no clinical implication are present in guidelines or expert’s opinion. Our study would add another confirmation of the SUVmax role in prediction of pancreatic cancer patients survival and it may be a start point for a literature review or a metanalysis that could define a median cut-off and usefulness of PET/CT in new prospective clinical studies.

Minor Comments:

Comment 1:

The authors should clearly state in abstract that patients with HIGH SUVmax have significantly WORSE survival compared to those with low SUVmax.

Replay:

  1. We added a new sentence in the abstract: (lines 34-35) we write. “Patients with a SUVmax ≤3.65 having significantly better survival than those with SUVmax >3.65 (p<0.001). ” than “(Patients with a SUVmax ≤3.65 having different survival curves from those with SUVmax >3.65 (p<0.001)”

Comment 2:

What kind of tomograph was used in 2009?

Replay:

  1. The first PET/CT tomograph( Biograph-16™, Siemens Healthcare GmbH) was in use from 2007 to 2012 and was upgraded from the manufacturer to a new version in 2009.We modify text of the paper (line 124), reporting that Biograph-16™, Siemens Healthcare GmbH was used from 2007 to 2012

Comment 3:

It is noted that there were 39 patients with R1 resection in main manuscript, whereas 38 patients in Table 1.

Replay

  1. We apologized but is another transcription error, the right value is 38 and right percentage (26.4%) (line 158)

Comment 4:

P values for serum CA 19-9 should be shown in Table 1.

Replay

  1. We add p values, because of the nature of CA 19-9 (continuous number) we reported mean and median; distribution in this cohort is not Normal so is best expressed by median, IRQ and its p-value. We added in Statistical analyses (line 147) t-student test.

We deleted “serum CA 19.9” from line 178 and added in lines 169-170   “Median values of CA 19-9,  numbers of patients with lymph node metastases ….”

Round 2

Reviewer 2 Report

The revised manuscript is appropriately corrected to the reviewer's comments.